# Berberine Ameliorates Metabolic-Associated Fatty Liver Disease Mediated Metabolism Disorder and Redox Homeostasis by Upregulating Clock Genes: Clock and Bmal1 Expressions

**DOI:** 10.3390/molecules28041874

**Published:** 2023-02-16

**Authors:** Cunsi Ye, Yajing Zhang, Shaomin Lin, Yi Chen, Zimiao Wang, Haoyinghua Feng, Guangqing Fang, Shijian Quan

**Affiliations:** School of Pharmaceutical Sciences, Guangzhou University of Chinese Medicine, Guangzhou 510006, China

**Keywords:** berberine, metabolic-associated fatty liver disease (MAFLD), insulin resistance, redox homeostasis, circadian misalignment

## Abstract

Metabolic-associated fatty liver disease (MAFLD) is one of the most common chronic liver diseases, which in turn triggers mild inflammation, metabolic dysfunction, fibrosis, and even cancer. Accumulating evidence has suggested that Berberine (BBR) could significantly improve MAFLD progression. Clock and Bmal1 as heterodimer proteins highly participated in the development of MAFLD, but whether BBR targets Clock and Bmal1 in MAFLD remains poorly understood. The result suggested that the protein levels of Clock and Bmal1 were decreased in MAFLD mice, which was negatively correlated with elevated reactive oxygen species (ROS) accumulation, the H_2_O_2_ level, liver inflammation, metabolic dysfunction, and insulin resistance. The mRNA and protein levels of Clock and Bmal1 were also decreased in glucosamine-induced HepG2 cells, which were are negatively related to glucose uptake, the ROS level, and the H_2_O_2_ level. More importantly, Bmal1 siRNA could mimic the effect of glucosamine in HepG2 cells. Interestingly, Berberine (BBR) could rescue metabolism disorder and redox homeostasis through enhancing Clock and Bmal1 expression in vivo and in vitro. Therefore, BBR might be an effective natural compound for alleviating redox homeostasis, metabolism disorder, and liver pathological changes in MAFLD by activating Clock and Bmal1 expression.

## 1. Introduction

Metabolic-associated fatty liver disease (MAFLD) is chronically hepatic steatosis with at least one out of the three metabolic risk factors (overweight/obesity, diabetes mellitus, or evidence of metabolic dysfunction) [1]. Worldwide, the estimated MAFLD prevalence was over 50% among overweight/obese adults [2]. MAFLD affects over a quarter of the global population, which in turn leads to severe complications including liver cirrhosis, liver cancer, and cardiovascular disease [3]. Insulin resistance (IR) is closely associated with many metabolic disorders, including hypertension, hyperlipidemia, atherosclerosis, and a fatty liver [4]. Increasing evidence suggests that IR plays critical roles in the progression of MAFLD through mediating glucose metabolism disorder and redox homeostasis [5,6]. However, the current molecular knowledge of IR on MAFLD remains poorly understood.

Previous studies suggested that the circadian rhythm is highly involved in the maintenance of metabolic homeostasis and redox homeostasis. The circadian rhythm is organized by the core circadian Clock genes, including circadian locomotor output cycles kaput (Clock), brain and muscle Arnt-like protein (Bmal1), Period (PER), and Cryptochrome (CRY) in the suprachiasmatic nucleus (SCN) of the brain [7]. These master regulators with the amplitude and/or phase of the rhythm were modulated in response to extracellular stimuli such as light and intracellular metabolic factors [8]. Chronic circadian disturbance and circadian rhythm sleep-wake disorders might induce glucose intolerance and hyperglycemia [9]. For example, it has been reported that the impairment of circadian rhythms leads to the development of diabetes via inducing abnormal insulin secretion, decreasing the insulin sensitivity, and exacerbating the inflammation [10]. Especially, when the circadian Clock component Bmal1 is disrupted, the subsequent endocrine adaptation has an important impact on insulin sensitivity, liver disease, and the metabolic syndrome [11,12]. The rhythmic expression of Bmal1 and Clock as master Clock-controlled genes are significantly involved in lipid metabolism and fatty liver disorder [13].

Many natural compounds from herbal medicine have been reported to correct metabolic disorders by affecting the circadian Clock genes’ expression [14]. For example, melatonin as a pineal gland hormone could improve glucose metabolism by down-regulating circadian rhythm expression [15]. Again, resveratrol could alleviate IR, reactive oxygen species (ROS) accumulation, and lipid metabolism in a high-fat diet mice model by reversing circadian misalignment [16,17]. In addition, nobiletin directly targets the orphan receptors related to retinoic acid receptors, indicating the possible mechanism of natural substances combating metabolic disease via the circadian gene network [18].

Berberine (BBR), as a well-known natural isoquinoline alkaloid, is derived from traditional Chinese herb plants including *Coptis chinensis*, *Berberis aquifolium*, *Berberis vulgaris*, and *Coscinium fenestratum* [19]. It has antioxidant, anti-inflammatory, anticarcinogenic, and anti-insulin resistant activities [20,21], and all of these capabilities participate extensively in oxidation dysfunction, metabolic disorders, and inflammation. Interestingly, BBR not only reduces the blood glucose level by stimulating cellular uptake [22], but it also improves intracellular insulin resistance and hepatic inflammation through the activation of AMPK and PI3K [23]. A recent study revealed that BBR might attenuate diet-induced obesity by modulating molecular Clock Bmal1 in BAT [24]. Our previous study also found that BBR might be very useful for the metabolic disorder treatments, ameliorating IR and regulating glucose and lipid metabolic dysregulation by enhancing the expression of GLUT4 proteins in the T2DM rats’ skeletal muscles [25]. However, the effect of BBR underlying the disrupted circadian rhythms on MAFLD remain to be fully elucidated.

In this study, we will fully investigate the regulatory roles of BBR on core Clock genes’ expression, relevant metabolism disorders, redox homeostasis, and MAFLD progression in vivo and in vitro.

## 2. Results

### 2.1. BBR Improves Insulin Resistance, Lipid Metabolism, and Hepatic Histology In Vivo

The circulating levels of fasting blood, insulin, Homa-IR, blood glucose, OGTT, ITT, TG, LDL, ALT, and AST were significantly increased in the MAFLD group (*p* < 0.05). Interestingly, the elevating of these blood indicators can be obviously alleviated by the BBR treatment (Figure 1, *p* < 0.05). Furthermore, the hepatic pathologies of steatosis, lobular inflammation, and hepatocyte ballooning were observed in the MAFLD group, which can be remarkably improved by the BBR treatment, as shown in Figure 2A. The NAS scores were increased in the MAFLD group and decreased in the MAFLD plus BBR group (Figure 2D, *p* < 0.05).

### 2.2. BBR Alleviated Glucosamine-Impaired Glucose Uptake in HepG2 Cells

To determine the effect of BBR on cell viability, CCK-8 was used to examine the cell growth of the HepG2 cells exposed to various concentrations of BBR (0–100 μM, respectively) for 18 h. As indicated in Figure 3A, the cell viability declined from 84.76 ± 6.13% to 60 ± 10.63% when the BBR concentration reached over 20 µM (*p* < 0.001). Therefore, we choose a BBR concentration ranging from 0 to 10 μM as the next functional study. Multiple cell models have demonstrated that glucosamine can produce insulin resistance and glucose intolerance [26], so glucosamine (20 mM) was applied to induce insulin resistance in HepG2 cells.

After the co-treatment with the glucosamine (20 mM) and BBR (1.25, 2.5, 5, and 10 μM) for 18 h, there was a significant decrease in the cell viability (*p* < 0.001). As depicted in Figure 3B, the cell survival percentage dropped linearly with the increase in the BBR concentration. After the administration of 10 μM BBR and 20 mM glucosamine, the cell viability was significantly decreased compared to that of the co-treatment with 10 μM BBR, and it was much lower than that in the group only treated with glucosamine (Figure 3B, *p* < 0.05). Consequently, BBR (1.25, 2.5, and 5 μM) was used for further glucose uptake experiments.

Figure 3C depicts the effect of different doses of BBR (1.25, 2.5, and 5 μM) with or without 20 mM glucosamine on glucose absorption in the HepG2 cells. It is clear that glucosamine exposure significantly decreased the glucose uptake capacity of HepG2 cells considerably from 100% to 42.56% (*p* < 0.001), indicating that it was a promising insulin-resistant model, whereas 18 h of co-treatment with BBR (1.25, 2.5, and 5.0 μM) enhanced the glucose uptake ability of glucosamine-treated HepG2 cells in a dose-dependent manner, and the glucose uptake was increased to 65.38%, 72.48%, and 83.77% (*p* < 0.05), respectively, suggesting that BBR might dose-dependently ameliorate glucosamine-induced IR in HepG2 cells.

### 2.3. BBR Alleviated Redox Imbalance In Vivo and In Vitro

The etiology of IR is intimately connected to cellular redox imbalance, including ROS signaling and oxidative stress [27]. Meanwhile, excessive ROS production may impose mitochondrial reductive stress and elicit multiple counterbalance metabolic, block signal transduction pathways, and interfere with normal physiological functions [28]. Excessive H_2_O_2_ emissions may impair the insulin signaling pathway, thereby leading to IR [29]. The deficiency of the circadian Clock gene Bmal1 may promote the accumulation of ROS, leading to dysregulation of the antioxidant defense pathway and glucose-lipid homeostasis [30,31].

As illustrated in Figure 4A, DCFH-DA was used to detect the redox state of HepG2 cells, and a substantial rise in the ROS level was detected in HepG2 cells subjected to the glucosamine treatment (raising from 100.00% to 320.67%, *p* < 0.001). After the co-treatment with 5 μM BBR, the ROS level decreased to 136%, indicating that BBR reduced the glucosamine-induced increase in ROS production. Similarly, a significant increase (from 98.76% to 173.21%) in H_2_O_2_ concentrations in the glucosamine-treated group was also detected, while it was decreased to 117.92% in the presence of BBR, showing that BBR effectively attenuated the glucosamine-induced H_2_O_2_ emission.

To determine whether the mRNA level of circadian Clock genes affects the effect of BBR on the redox state, ARNTL-targeting small interfering RNA (siRNA) (si-Bmal1) was designed and used to decrease the Bmal1 protein abundance. As demonstrated in Figure 4A,B, the relative ROS level was increased from 141.01% to 202.54% when Bmal1 was knocked down, suggesting that the knockdown of Bmal1 gene diminished the inhibitory impact of BBR on ROS formation. The H_2_O_2_ level exhibited the similar trend, it increased from 112.92% in the co-treatment group to 143.61% in the si-Bmal1 group (Figure 4C). Importantly, the ROS and H_2_O_2_ levels were increased in the MAFLD mice and decreased in the MAFLD mice treated with BBR (*p* < 0.05) in Figure 2E,G. This indicates that BBR plays an inhibitory role in glucosamine-induced H_2_O_2_ emission by promoting Bmal1 expression.

### 2.4. BBR Upregulated Clock/Bmal1 Expression In Vivo and In Vitro

It has been observed that abnormalities in circadian rhythm genes are directly connected with metabolic disruption [32,33]. The protein-to-protein interaction network for Clock, Bmal1 (ARNTL), and other circadian rhythm genes from STRING protein database (https://string-db.org/, accessed on 6 September 2022) is shown in Figure 3F. The hepatic expression Clock and Bmal1 proteins by Western blot were remarkably decreased in the MAFLD group, which can be significantly increased with the BBR treatment (*p* < 0.05) in Figure 2I,J. Meanwhile, the IHC images in Figure 2B,C,F,H also confirm the expression of Clock and Bmal1 proteins in the MAFLD and MAFLD plus BBR group.

Exposure to glucosamine or the BBR treatment are related to the transcription of Clock and Bmal1 under insulin-resistant conditions in HepG2 cells, as illustrated in Figure 3C,D. Intriguingly, the mRNA expression levels of Clock and Bmal1 revealed comparable cyclic oscillations, with both of the levels reaching their lowest points at ZT24 or ZT30. According to Figure 5A,B, consistent with the alteration in the mRNA levels, the involvement of glucosamine caused relatively shallow circadian oscillations, and the circadian amplitude was decreased. Compared with the control group, the protein expression of Bmal1 in the glucosamine co-treatment group tends to have a reverse-circadian phase in antiphase with that of control. The expression appeared to be greatest at ZT24, while other time points had a relatively low expression. Although the amplitude of Bmal1 cycling was low, the protein expression rhythm in the BBR treatment group was similar to that of the controls. Moreover, the BBR co-treatment more remarkably reversed their phase compared with that of the glucosamine group. The BBR co-treatment successfully restored the oscillatory behavior to be similar to their transcript levels. Overall, in the glucosamine-induced insulin-resistant HepG2 cells model, BBR regulates circadian Clock expression significantly at the protein and RNA levels.

### 2.5. BBR Alleviated Glucose Metabolism Disorder in a Bmal1-Dependent Manner

To investigate the function of Bmal1 in rhythmicity controlling circadian gene and glucose metabolism, we knocked down Bmal1 by using its siRNA, and the relative protein levels of major circadian Clock genes and glucose uptake were assessed. The Western blots show that si-RNA knockdown of Bmal1 in HepG2 cells dramatically decreased its protein level to 48.37% (Figure 5D). In addition, the relative protein expressions of Clock were also decreased to 67.32% compared with that of the control group, indicating that Bmal1 is the core Clock gene associated with Clock to form a Clock/Bmal1 heterodimer (via E-box elements) and is likely to contribute to the remodeling of circadian gene expression involved in establishing circadian patterns, which is in keeping with earlier reports [34]. Moreover, the down-regulation of Bmal1 and Clock was detected in HepG2 cells induced by glucosamine, whereas Bmal1 and Clock down-regulation were reversed by the co-treatment with glucosamine and BBR (Figure 5D,E). However, the protein levels in the BBR co-treatment group were not significantly increased after Bmal1 knockdown (*p* > 0.05), indicating that the role of BBR on the expression of circadian rhythm may be dependent on the cyclic expression of Bmal1.

Next, the glucose uptake ability of HepG2 cells after the silence of Bmal1 by siRNA was determined to evaluate the effect on glucose metabolism. Compared to that of the controls, Bmal1 knockdown inhibited the rate of glucose uptake to 71.53%, showing that the inhibition of glucose metabolism could be related to the knockdown of Bmal1. As demonstrated in Figure 5F, although BBR raised the relative glucose absorption level from 51.5% to 81.36% in comparison to that of the glucosamine-treated group, Bmal1 siRNA reversed this effect dramatically. The glucose uptake was only at 43.6% in the BBR-co-treated Bmal1 knocked-down group, therefore, it can be concluded that the knockdown of Bmal1 blocks BBR-mediated glucose uptake and BBR regulated glucose metabolism in a Bmal1-dependent manner under glucosamine-induced condition.

## 3. Discussion

In this study, the potential mechanisms underlying BBR’s effects were to improve MAFLD-induced circadian misalignment, redox imbalance, metabolism disorder, and insulin resistance. The main findings were the following ones: (i) the BBR treatment could impair liver steatosis, lobular inflammation, hepatocyte ballooning, and NAS score; (ii) BBR could enhance the expression of Clock and Bmal1; (iii) BBR mitigated the redox imbalance mediated liver inflammation by decreasing the production of ROS and H_2_O_2_; (iv) the glucose metabolism regulated by BBR was depending on the cyclic expression of Bmal1 gene. All of these suggest that the natural product BBR might promote glucose metabolism, redox balance, and regular circadian rhythm by activating the core circadian Clock gene Bmal1.

Circadian clocks are self-sustaining oscillators that coordinate our daily physiological behavior due to environmental changes [35]. The suprachiasmatic nucleus (SCN) is the principal pacemaker responsible for synchronizing internal circadian rhythms with the external day–night cycle through humoral factors or autonomic innervation, governing the circadian rhythms of physiology and behavior [36]. It is known that transcriptional and post-transcriptional feedback loops involving a group of circadian proteins govern circadian rhythms. In mammals, the endogenous clock system consists of a well-characterized transcriptional and translational feedback loop (TTFL) composed of core clock gene elements: heterodimers composed of transcription factor activators (Clock and Bmal1) proteins that activate the transcription of inhibitor genes (PERs and CRYs) [37]. The transcriptional activation of circadian genes by Bmal1-Clock heterodimers can bind to the E-box of CACGTG, and this results in the expression of PERs, CRYs, and other related genes [38]. Multiple health morbidities are caused by the misalignment of biological and behavioral circadian rhythms, including metabolic syndrome [39], obesity [40], and type II diabetes [41]. Insulin sensitivity is a vital indicator of metabolic health, since cellular insulin resistance is increasingly recognized as a key contributor to the development of metabolic syndrome and type 2 diabetes. It was discovered that participants treated with forced circadian misalignment (a mimic of shift work or jet lag) exhibited impaired glucose tolerance, insulin resistance, and elevated levels of oxidative stress [42]. Bmal1-deficient diabetic rats suffered from circadian rhythm disturbance, β-cell secretion abnormality, hyperglycemia, and impaired glucose tolerance [41]. The genetic deletion of Bmal1 or Clock reduces the insulin-stimulated glucose uptake and the phosphorylation of InsR, AKT, and GSK3 [43]. Glucosamine increases insulin resistance, which is ameliorated by capsaicin in a Bmal1-dependent manner [44]. Consistent with previous studies, the mRNA and protein expression of Clock and Bmal1 were related to rhythmic oscillations in the control group. Mounting evidence from pharmacological and genetic studies has demonstrated that Clock-enhancing strategies that modulate circadian amplitude are promising approaches for the treatment of metabolic diseases and other Clock-related systemic pathologies [39,45]. Hence, there is growing interest in searching for natural products against metabolic syndrome, especially those targeting circadian rhythm-related proteins.

BBR, a bio-active phytoconstituent mainly extracted from *Coptis chinesis*, has been used for centuries in traditional Chinese medicine to treat metabolic diseases such as diabetes and obesity [46]. It is well known that BBR regulates glycolipid metabolism, contains anti-inflammatory and anticytotoxic effects, and decreases insulin resistance. Our previous study showed that BBR could ameliorate insulin resistance in mice by reducing inflammation and promoting the expression of insulin signaling GLUT4 [25]. In addition, BBR was discovered to have anti-diabetic benefits via activating AMPK, reducing HOMA-IR value and insulin resistance [47], and enhancing insulin resistance and boosting glucose intake in 3T3-L1 mouse pre-adipocytes [48]. Through the promotion of FGF21 production by Bmal1, altering the level of molecular Clock and Bmal1 in BAT can indeed lessen the effects of disorders associated with circadian rhythm [24]. However, it remains unclear if BBR reduces insulin resistance in part via regulating the circadian rhythm. In our study, an in vitro insulin-resistant cell (IR-C) model was established using HepG2 cells to evaluate the BBR’s effects on glucose uptake and metabolism, representing a decrease in hepatic gluconeogenesis inhibition. As is well known, in T2DM, low glycogen synthesis and high gluconeogenesis and glycogenolysis are important mechanisms responsible for fasting hyperglycemia [49]. Compared with that of the control, a significant reduction of the glucose uptake (about 60%) was observed in glucosamine treatment group, confirming that insulin resistance was successfully induced by the glucosamine treatment in HepG2 cells, and the signaling pathways involved in glucose uptake were impaired by the addition of glucosamine. BBR and glucosamine co-treatment further reverted the effect of insulin resistance induced by glucosamine by 83.77% in comparison with that of the glucosamine group, demonstrating that BBR could improve glucosamine-stimulated glucose uptake and intracellular glucose metabolism. However, the glucose uptake significantly declined once the expression of Bmal1 was inhibited, implying the circadian Clock gene Bmal1 likely plays an important role in regulating the daily rhythm of glucose metabolism and glucose homeostasis.

Circadian rhythms are presented in several interconnected activities such as metabolism and oxidation-reduction cycles [50]. Emerging studies suggest that circadian clocks maintain physiological levels of ROS and protect organisms from oxidative stress via antioxidants and antioxidant enzymes expressed in circadian rhythms [50]. The overproduction of ROS directly activates serine/threonine kinase cascades, which increases the serine phosphorylation of IRS and lowers its capacity to undergo tyrosine phosphorylation, resulting in decreased glucose absorption in the liver. In addition, increased ROS levels may interfere with insulin signaling and lead to insulin resistance [51]. In the present study, the intracellular ROS and H_2_O_2_ levels were significantly increased in the glucosamine-induced HepG2 cells and liver tissues of MAFLD mice, implying an elevated oxidative metabolism. However, the BBR co-treatment remarkably reduced the ROS levels, leading to the inductively enhanced antioxidant capacity of HepG2 cells. After the Bmal1 knockdown, the increased levels of ROS and H_2_O_2_ suggested a diminished capability of BBR to manage the cellular redox balance of HepG2 cells. More importantly, the antioxidant protective effects of BBR might be mediated by the normal expression of the core circadian gene Bmal1.

Consistent with the previous study [44], our results showed that the glucosamine treatment could not only reduce glucose uptake and insulin resistance, but also decrease the oscillation amplitude of the Clock gene (Clock and Bmal1) and change the oscillation’s phase. Given the bidirectional nature between glucosamine-induced circadian disruption and metabolic pathologies, circadian disruption may in turn lead to a vicious cycle with increased ROS generation and contribute to the progression and worsening of metabolic diseases and circadian rhythm disturbances. Interestingly, BBR could improve the glucose intake of HepG2 cells and mitigate glucosamine-induced circadian rhythm disorders and the damage induced by oxidative stress. Moreover, the knockdown of Bmal1 significantly attenuated the protective effect of BBR on glucosamine-mediated ROS accumulation and glucose uptake.

In MAFLD mice, we found that Berberine can alleviate hepatic redox imbalances by reducing the levels of ROS and H_2_O_2,_ which are negatively corelated with liver lobular inflammation and inflammatory cytokines. Previous research has shown that an imbalance in the redox state can directly will trigger inflammation and metabolic reprogramming in the liver [52]. Our results suggest that Berberine can reduce the release of inflammatory cytokines and liver steatosis of liver in mice with MAFLD by improving hepatic oxidative stress.

## 4. Materials and Methods

### 4.1. Materials and Reagents

Berberine (B3251) (purity > 98%) and dimethyl sulfoxide (DMSO) were purchased from Sigma-Aldrich (St. Louis, MO, USA). Glucosamine (purity > 99%) was purchased from the Beyotime Biotechnology Company (Nanjing, China). The Roswell Park Memorial Institute 1640 medium (RPMI-1640) growth medium was purchased from (Hyclone Co., Logan, UT, USA). Fetal bovine serum (FBS), penicillin, and streptomycin were purchased from Lonza Walkerrsville (GIBCO CO., Allendale, NJ, USA). The concentrations of ROS and H_2_O_2_ in the liver were measured using a mouse ROS ELISA kit (RD-RX20367-48T) and H_2_O_2_ ELISA kit (RD-RX21392-48T) (Beijing Ruida Henghui Technology Development Co., Ltd., Beijing, China). HepG2 cells were purchased from the American Type Culture Collection (ATCC). Antibodies against Clock (ab93804) and Bmal1 (ab93806) were purchased from Abcam (Abcam, Cambridge, MA, USA). The cell counting kit-8 (CCK8) assay kit was purchased from DoJinDo (DoJinDo ChemTech Lim, Kumamoto, Tokyo, Japan). The Glucose Oxidase Assay Kit was obtained from APPLYGEN (APPLYGEN, Beijing, China). The cellular ROS and H_2_O_2_ assay kit was purchased from Beyotime (Biotechnology, Nanjing, China). Machine-filtered, pure water (Milli-Q) was used.

### 4.2. MAFLD Model and Treatment

All of the animal experiments were performed by referring to the animal ethics, which were approved by the Animal Care Committee of Guangzhou University of Chinese Medicine. Twenty-seven six-week-old male C57BL/6 mice were purchased from the Animal Center of the Guangzhou University of Chinese Medicine and maintained under controlled light conditions (12:12 light–dark cycle), with a normal chow diet and water provided ad libitum. After, the mice were acclimatized for 7 days, and 27 individuals were divided into 3 groups: (1) normal healthy individuals of the C57BL/6 mice as a positive control group (*n* = 9); (2) MAFLD group (*n* = 9); (3) MAFLD plus BBR group (*n* = 9). The MAFLD mice were received daily berberine. The normal control group was fed a control diet, the MAFLD group was fed high-fat diet for 12 weeks, and the MAFLD plus BBR group was firstly given high-fat diet for 8 weeks, and then given BBR (200 mg/kg/d by gavage) for 4 weeks. The high-fat diet containing 60% calories from fat (D12492) was purchased from Xietong Pharmaceutical Bio-engineering Co., Ltd. (Nanjing, China). The blood and liver tissue were stored for ELISA, Western blot, and H&E staining.

### 4.3. Serum Indexes

The fasting blood glucose levels were tested with Accu-chek Inform II strips (Roche, Mannheim, Germany) using a glucometer Accu-chek Performa (Roche, Indianapolis, Indiana, USA) after 4 weeks of BBR treatment for the Oral glucose tolerance test (OGTT) and Insulin tolerance test (ITT). The levels of triglyceride (TG), low-density lipoprotein cholesterol (LDL), Alanine/aspartate aminotransferase (ALT/AST), and insulin were tested using the ELISA kit (TG Cat. A113-1-1; LDL Cat. H207-96t; ALT Cat. C009-2-1; AST Cat. C010-2-1) from Nanjing Jiancheng Bioengineering Institute (Nanjing, China), and the insulin Cat. SEKM-0141 from Solarbio Biotech. Company (Beijing, China).

### 4.4. Hepatic ROS and H_2_O_2_ Levels by ELISA

The hepatic ROS and H_2_O_2_ content were determined using the mouse ROS ELISA kit (RD-RX20367-48T) and H_2_O_2_ ELISA kit (RD-RX21392-48T) (Beijing Ruida Henghui Technology Development Co., Ltd., Beijing, China) according to the manufacturer’s instructions. The HOMA-IR score was calculated using the HOMA-IR formula (HOMA-IR = fasting insulin (mU/L) × fasting glucose (mmol/L)/22.5 [53]).

### 4.5. H&E Staining

Hematoxylin and eosin stain (H&E staining) was used for liver histology according to instructions from a previous study [54]. The NAS score ranging from 0 to 8 was used to evaluate the scores of liver steatosis (0–3), lobular inflammation (0–3), and hepatocyte ballooning (0–2).

### 4.6. Cell Culture and Treatment

HepG2 cells were grown in RPMI1640 medium (Hyclone Co., Logan, UT, USA) supplemented with 10% FBS and 1% penicillin/streptomycin (GIBCO CO., Allendale, NJ, USA) at 37 °C in a humidified atmosphere of 5% CO_2_. Various concentrations of BBR were added to HepG2 cells cultured in RPMI1640 medium supplemented with 0 mM or 20 mM glucosamine and cultured for another 18 h. Then, at 6 h intervals between 18 and 42 h, cells were collected for subsequent quantification of mRNA and protein expression.

### 4.7. siRNAs Transfection of HepG2 Cells

HepG2 cells were seeded in 6-well plates, then the si-Ctrl or si-Bmal1 plasmids were transfected into the cells using liposomes by using a modified protocol for Lipofectamine 2000 (Invitrogen), following the manufacturers. After 48 h transfection, the culture media were supplemented with 20 mM glucosamine and 5 µM BBR for 18 h.

### 4.8. CCK-8 Cell Viability Analysis

The effect of BBR on HepG2 cell viability was determined by a CCK-8 assay. The cells were seeded into 96-well plates at a density of 2 × 10^4^ cells/well maintained in RPMI1640 medium supplemented with 10% FBS. Following overnight incubation, the cells were treated with different concentrations of BBR (0, 1.25, 2.5, 5.0, 10, 20, 30, 40, 60, 80, and 100 μM) with or without glucosamine (20 mM) for 18 h. After being treated, 90 μL of medium and 10 μL of CCK-8 solution were added to each well and incubated for 2 h at 37 °C. Optical density (OD) values were measured at 450 nm using a microplate reader (Bio-Rad, Hercules, CA, USA). Cell viability was standardized to that of the untreated controls.

### 4.9. Glucose Uptake Assay

As previously disclosed, the insulin resistance model was established by treating HepG2 cells with 20 mM glucosamine for 18 h with or without BBR (1.25, 2.5, 5.0, and 10 µM). The residual glucose concentration in the culture supernatant was measured with the glucose oxidase (GOD) method and determined at 550 nm using a glucose assay kit (Applygen, Beijing, China). The glucose content of every experimental group with 5 parallel wells settled and the original RPMI1640 media was measured. The glucose content in the samples was calculated based on the standard curve of glucose concentration vs. optical density. Glucose uptake was calculated based on the difference between the glucose content of the final cell supernatant and that of the original medium [26]. The glucose uptake rates (%) = Glucose content subtraction (experimental groups)/Glucose content subtraction (control groups) × 100%.

### 4.10. Measurement of Intracellular ROS Generation and H_2_O_2_ Level

The ROS assay kit (Beyotime Biotechnology, Nanjing, China) was used to determine intracellular and hepatic ROS accumulation according to the manufacturer’s instructions. The HepG2 cells were cultivated in a 6-well plate and exposed to the various treatments mentioned in the preceding section, and then we collected the samples. After two washes with cold PBS, the cells were treated for 20 min in the dark at 37 °C with DCFH-DA. Then, fluorescence microscopy (488 nm filter, OLYMPUS IX-71, Olympus Corporation, Tokyo, Japan) was used to record the fluorescent signal. The fluorescence intensity was measured, and the amount of ROS was evaluated using Image J software to assess the fluorescence intensity. The experiment was performed in triplicates and repeated three times.

For H_2_O_2_ detection, the HepG2 cells were co-treated with different doses of BBR and 20 mM glucosamine for 18 h before being lysed in 100 µL lysis buffer. Fresh liver tissue was cut off and placed into DEME medium for cell culture for 24 h. Then, the supernatants collected after 4 min of centrifugation at 12,000× *g* were utilized to assess the intracellular H_2_O_2_ levels using a commercial Kit (Beyotime Institute of Biotechnology, Nanjing, China). The sample solution (50 µL) was incubated with the reaction solution (100 µL) at room temperature for 30 min. Finally, the absorbance at 560 nm was measured using microplate reader (SpectraMax i3X, Molecular Devices, San Jose, CA, USA).

### 4.11. RNA Extraction, cDNA Synthesis, and Real-Time PCR Analysis

According to the manufacturer’s instructions, the total RNA was extracted from HepG2 cells using the Cell RNA Purification Kit (EZBioscience Catalog # EZB-RN001-plus, Roseville, CA, USA). The relative mRNA quantification was assessed using RT-qPCR as described [55]. The primer sequence is in Table 1. The relative expression level of target genes was measured using 2^−ΔΔCT^ method and standardized to glyceraldehyde-3-phosphate dehydrogenase (GAPDH).

### 4.12. Western Blot Analysis

The HepG2 cells and mice liver tissues were collected and lysed in RIPA lysis buffer (Beyotime, Nanjing, China) for utilizing protein concentration. The standard Western blot protocol was conducted as reported [44]. The optical densities of Clock, Bmal1, and GAPDH were calculated using Image J.

### 4.13. Statistical Analysis

Data are expressed as means ± standard deviation. One-way analysis of variance (ANOVA) was used to compare the difference among three groups with LSD method for multiple comparisons. All of the analyses were conducted using SPSS version 22.0 software, and *p* < 0.05 indicates a statistical significance.

## 5. Conclusions

Overall, our findings support the role of Berberine in mitigating glucose metabolism, insulin resistance, lipid dysfunction, and liver inflammation in the progression of MAFLD by Clock genes Bmal1/Clock-mediated oxidative stress and circadian misalignment. BBR might be an effective potential natural compound for MAFLD treatment by targeting Bmal1/Clock protein heterodimers and its antioxidant property. Further analysis of BBR and Bmal1/Clock heterodimers interaction will be further clarified in the future.

## Figures and Tables

**Figure 1 molecules-28-01874-f001:**
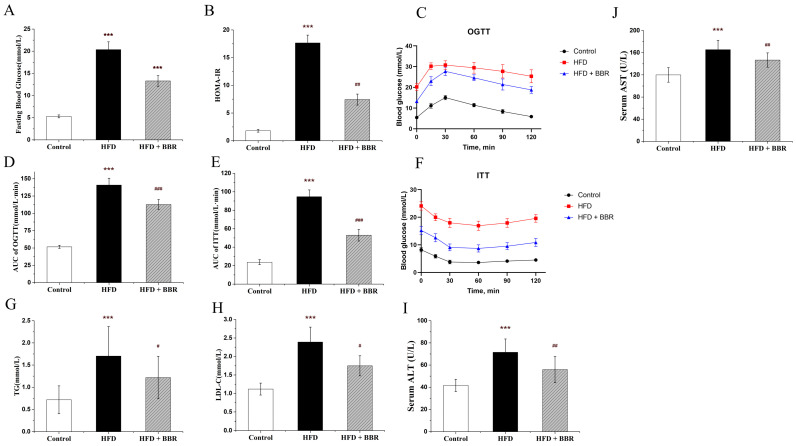
BBR improved insulin resistance and lipid disorder in vivo. (**A**) Fasting blood insulin level; (**B**) Homa-IR; (**C**,**D**) oral glucose tolerance test (OGTT); (**E**,**F**) insulin tolerance test (ITT); (**G**) triglyceride (TG); (**H**) low-density lipoprotein (LDL); (**I**) serum ALT level; (**J**) serum AST level. Data are presented as the mean value ± SE (*n* ≥ 6): *** *p* < 0.001 compared to the control group. ^#^ *p* < 0.05, ^##^ *p* < 0.01, and ^###^ *p* < 0.001 compared to the HFD group.

**Figure 2 molecules-28-01874-f002:**
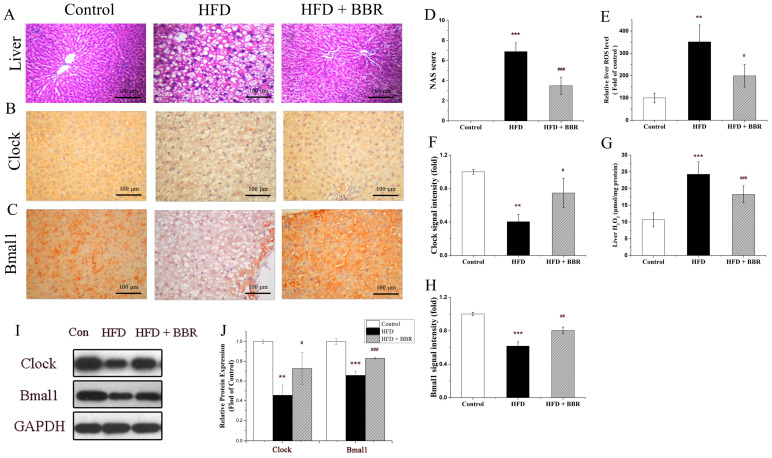
BBR improved liver pathology, ROS level, H_2_O_2_ level, and enhanced the expression of Clock and Bmal1 proteins in vivo. (**A**) Liver H&E staining; (**B**) IHC of Clock; (**C**) IHC of Bmal1; (**D**) liver NAS score; (**E**) liver ROS level; (**F**) relative Clock protein expression by IHC; (**G**) liver H_2_O_2_ level; (**H**) relative Bmal1 protein expression by IHC; (**I**) Clock, Bmal1, and GAPDH protein bands; (**J**) relative Clock and Bmal1 protein expression by Western blot. Data are presented as the mean value ± SE (*n* ≥ 6): ** *p* < 0.01, and *** *p* < 0.001 compared to the control group. ^#^ *p* < 0.05, ^##^ *p* < 0.01, and ^###^ *p* < 0.001 compared to the HFD group.

**Figure 3 molecules-28-01874-f003:**
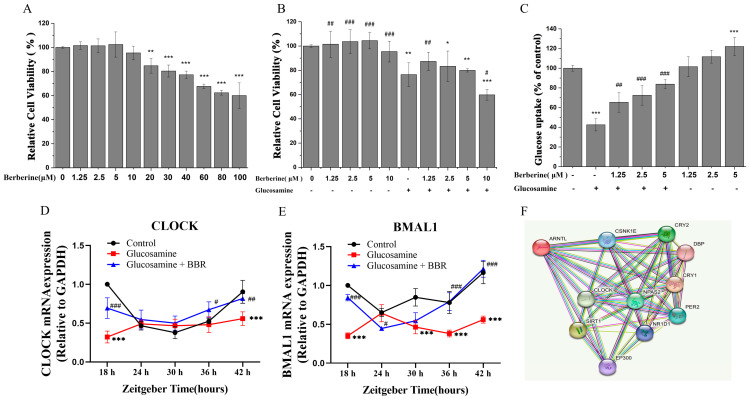
BBR alleviated glucosamine-impaired glucose uptake and increased mRNA of Clock and Bmal1 in HepG2 cells. (**A**) Relative viability of HepG2 cells treated with different concentrations of BBR measured by CCK-8 assay; (**B**) relative cell viability treated with BBR (1.25, 2.5, 5.0, and 10.0 μM) and co-treated with/without 20 mM glucosamine; (**C**) glucose uptake in groups treated with BBR (1.25, 2.5, and 5.0 μM) and co-treated with/without glucosamine (20 mM); (**D**) mRNA expression of Clock; (**E**) mRNA expression of Bmal1; (**F**) protein-to-protein interaction of Clock and Bmal1. Data are presented as the mean value ± SE (*n* ≥ 6): * *p* < 0.05, ** *p* < 0.01, and *** *p* < 0.001 compared to the control group. ^#^ *p* < 0.05, ^##^ *p* < 0.01, and ^###^ *p* < 0.001 compared to the glucosamine group.

**Figure 4 molecules-28-01874-f004:**
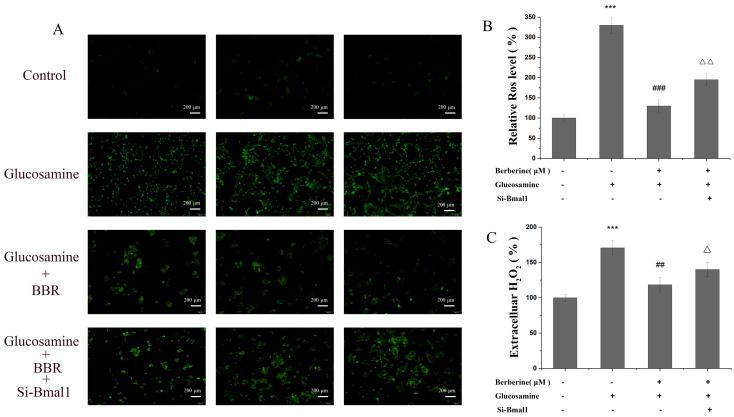
BBR reversed glucosamine-induced redox imbalance in HepG2 cells. (**A**,**B**) The status of ROS level in different groups was detected using DCFH-DA to examine the cellular oxidation status. (**C**) Production of H_2_O_2_ measured by a commercial kit. All data are presented as the mean ± SEM (*n* = 3). *** *p* < 0.001 compared to the control group; ^##^ *p* < 0.01, and ^###^ *p* < 0.001 compared to the glucosamine group; ∆ *p* < 0.05, and ∆∆ *p* < 0.01 compared to the glucosamine plus BBR group.

**Figure 5 molecules-28-01874-f005:**
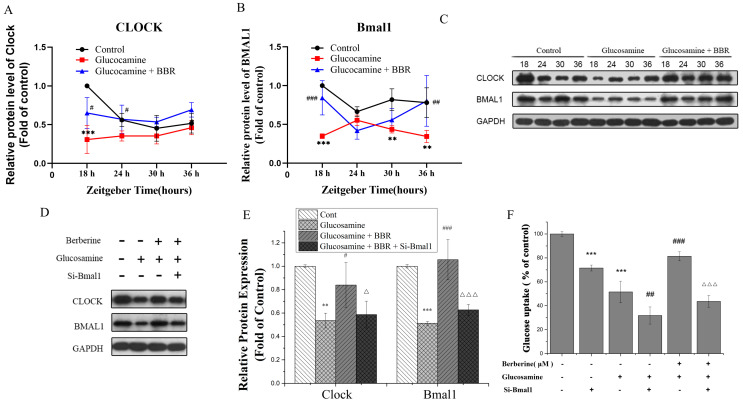
BBR enhanced the expressions of Clock and Bmal1 after glucosamine or Bmal1 siRNA treatment. (**A**–**C**) Relative expression of Clock and Bmal1 proteins, as well as WB images of Clock and Bmal1 after transfection and treatment with glucosamine and BBR in HepG2 cells based on different time. (**D**–**F**) Relative expression of Clock and Bmal1 proteins, as well as WB images of Clock and Bmal1 after treated with si-Bmal1, BBR (5 μM), and glucosamine for 48 h. All data are presented as the mean ± SEM (*n* = 3). ** *p* < 0.01, and *** *p* < 0.001 compared to the control group; ^#^ *p* < 0.05, ^##^ *p* < 0.01, and ^###^*p* < 0.001 compared to the glucosamine group; ∆ *p* < 0.05, and ∆∆∆ *p* < 0.001, compared to the glucosamine plus BBR group.

**Table 1 molecules-28-01874-t001:** Primer sequences used for quantitative real-time PCR analysis.

	Forward Primer	Reverse Primer
GAPDH	5′-ACCCAGAAGACTGTGGATGG-3′	5′-ACCCAGAAGACTGTGGATGG-3′
CLOCK	5′-ACAGGGCACCACCCATAATA-3′	5′-TCCACTGTTGCCCCTTAGTC-3′
BMAL1	5′-GTAACCTCAGCTGCCTCGTC-3′	5′-AGCTGTTGCCCTCTGGTCT-3′
PER1	5′-CAATGGTTCAAGTGGCAATG-3′	5′-TGTAGGCAATGGAACTGCTG-3′
PER2	5′-CCGGAGTTAGAGATGGTGGA-3′	5′-AGTAATGGCAGTGGGACTGG-3′
CRY1	5′-GTGTTTCCCAGGCTTTTCAA-3′	5′-TGGTTCCATTTTGCTGATGA-3′
CRY2	5′-TACCTGCCCAAATTGAAAGC-3′	5′-GCGAAAGCTGCTGGTAAATC-3′

## Data Availability

The raw data supporting the conclusions of this article will be made available by the authors, without undue reservation.

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
