# Peer review of "Berberine Ameliorates Metabolic-Associated Fatty Liver Disease Mediated Metabolism Disorder and Redox Homeostasis by Upregulating Clock Genes: Clock and Bmal1 Expressions"

_molecules, 2023, doi:10.3390/molecules28041874_

Round 1

Reviewer 1 Report

This manuscript reported berberine could ameliorate the metabolic associated fatty liver disease, and demonstrated the potential regulation of CLOCK and BMAL1. However, there are some concerns need to be addressed before acceptance.

1. The manuscript was poor prepared, the resolution of Figures in main text was too low. The authors should provide the high resolution images. And some sentences need English polish, such as figure 1 legend, should be revise as "BBR improves blood insulin resistance and suppress lipid accumulation".

2. In Figure 2C, the ROS level by DCFH reagent, the emission fluorescence intensity could comprehensively reflect the ROS level, other than the only one Em point. Furthermore, as a fluorescence probe, I recommend the authors include fluorescence imaging of cells by confocal imaging.

3. In Figure 2E, the CLOCK and BMAL1 were measured by Western blotting, I also recommend the inclusion of IHC images in this panel, which could further confirm the association of CLOCK and BMAL1 in BBR-mediated HFD.

4. In Figure 3, the authors showed BBR increased the mRNA level of CLOCK and BAML1, did BBR regulate the two molecules by regulating its transcriptional level or protein level or both? I recommend the author supply the verification by CHX、MG132 (protein level) or luciferase reporter assay (transcriptional level).

5. Some author affiliation information is missing. 

Author Response

We appreciate your valuable comments.

Comment 1.1. The manuscript was poor prepared, the resolution of Figures in main text was too low. The authors should provide the high resolution images. And some sentences need English polish, such as figure 1 legend, should be revise as "BBR improves blood insulin resistance and suppress lipid accumulation".

Response 1.1: The manuscript was thoroughly revised by us, and a native speaker has polished language. The resolution of Figures has been obviously improved. In addition, the description of Figure 1 legend of "BBR improves blood insulin resistance and suppress lipid accumulation" was changed into “BBR improved insulin resistance and lipid disorder in vivo

Comment 1.2. In Figure 2C, the ROS level by DCFH reagent, the emission fluorescence intensity could comprehensively reflect the ROS level, other than the only one Em point. Furthermore, as a fluorescence probe, I recommend the authors include fluorescence imaging of cells by confocal imaging.

Response 1.2: In Figure 2C, the hepatic content of ROS was detected by ELISA in our study. Therefore, we don’t have fluorescence images of ROS. The ROS ELISA methodology was on Page 3 (Red color marked).

Comment 1.3. In Figure 2E, the CLOCK and BMAL1 were measured by Western blotting, I also recommend the inclusion of IHC images in this panel, which could further confirm the association of CLOCK and BMAL1 in BBR-mediated HFD.

Response 1.3: IHC images of CLOCK and BMAL1 have been added in Figure 2B and 2C, which were consistent with the trend of CLOCK and BMAL1 by Western blotting.

Comment 1.4. In Figure 3, the authors showed BBR increased the mRNA level of CLOCK and BAML1, did BBR regulate the two molecules by regulating its transcriptional level or protein level, or both? I recommend the author supply the verification by CHX、MG132 (protein level) or luciferase reporter assay (transcriptional level).

Response 1.4: As we mainly focus on the expression and function of CLOCK and BAML1 gene by BBR, we previously haven’t considered the effect of CHX and MG 132. The time for revision was limited, therefore, CHX、MG132 (protein level) or luciferase reporter assay (transcriptional level) will be further studied in the future.

Comment 1.5. Some author affiliation information is missing. 

Response 1.5: Author affiliation information has been updated.

Reviewer 2 Report

1.     As it said in Abstract, did authors check mild inflammation and metabolic dysfunction in 12 weeks HFD fed HFD mice as it is said to induce MAFLD

2.     Did authors check liver damage marker such as aspartate transaminase (AST) and alanine transaminase, aka alanine aminotransferase (ALT) in the serum level of mice MAFLD mice?

3.     Only showing the liver morphology by H&E staining doesn’t consider it as liver. Author need to showed more evidence

4.     How author mimic MAFLD in HepG2 cells, just by treating the cell with glucosamine?

5.     In metarials and method section, how BBR is treated in mice? Is it injected or bBR is given to food? Please elaborate it

6.     What is meaning of standard protocol? Is it universally accepted protocol ? please elaborate more in materials and method section

Author Response

We are appreciated your valuable comments

Comment 1.1: As it said in Abstract, did authors check mild inflammation and metabolic dysfunction in 12 weeks HFD fed HFD mice as it is said to induce MAFLD

Response 1.1: We confirmed the mild inflammation and metabolic dysfunction by liver pathology and blood biochemistry indexes such as blood insulin, blood glucose, OGTT, ITT, TG, LDL, ALT, and AST.

Comment 1. 2: Did authors check liver damage markers such as aspartate transaminase (AST) and alanine transaminase, aka alanine aminotransferase (ALT) in the serum level of mice MAFLD mice?

Response 1.2: The ALT and AST level have been detected in the serum level of mice MAFLD mice in Figure 1.

Comment 1.3: Only showing the liver morphology by H&E staining doesn’t consider it as liver. Author needs to showed more evidence

Response 1.3: Liver morphology by H&E staining is not enough. Therefore, we included the NAS score for liver morphology and blood biochemistry indexes, which can comprehensively support our conclusion.

Comment 1.4: How author mimic MAFLD in HepG2 cells, just by treating the cell with glucosamine?

Response 1.4: Previous studies widely accept the glucosamine-induced cell model for hepatocyte insulin resistance, oxidative stress, and circadian clock genes  (Yan J, et al. Pharmacol Res. 2018. PMID: 29284152; Mi Y, et al. Mol Nutr Food Res. 2017. PMID: 28869341). Meanwhile, we found insulin resistance through decreased glucose uptake and oxidative stress by increased ROS and H2O2 levels in glucosamine-induced HepG2 cells. Insulin resistance and oxidative stress play an important role in MAFLD progression.

Comment 1.5: In metarials and method section, how BBR is treated in mice? Is it injected or BBR is given to food? Please elaborate it

Response 1.5: MAFLD mice were given by gavage (BBR 200 mg/kg/d) daily for 4 weeks, which was supplied in the material and method section

Comment 1.6: What is the meaning of standard protocol? Is it a universally accepted protocol? please elaborate more in the materials and method section

Response 1.6: Standard protocol means experimental conduction according to reagent instructions. The detailed description was elaborated in the materials and method section.

Round 2

Reviewer 1 Report

The authors have addressed some of my concerns, however, there are some concerns about their manuscript.

1. The authors claimed that the liver ROS was examined by DCFH-DA in methods section, DCFH-DA is a fluorescence probe, the authors still claimed they examined through ELISA, the authors did not understand my concerns.

2. Some language errors are still not revised in the revised version, such as  "DCFDA" in the main context and figure legend should be revised as "DCFH-DA"

3. In revised Figure 2, the scar bar of IHC image is missing. And the related problem was also occurred in Figure 4.

Author Response

Comment 1.1. 1. The authors claimed that the liver ROS was examined by DCFH-DA in methods section, DCFH-DA is a fluorescence probe, the authors still claimed they examined through ELISA, the authors did not understand my concerns.

Response 1.1: We sincerely apologize for the misunderstanding regarding the previous comment on confocal imaging. You are correct that using multiple emission points in confocal imaging can provide a more detailed visualization of the ROS content using DCFH-DA staining and the cells' nuclei using DAPI staining. Unfortunately, our university's confocal microscope is currently undergoing repairs, but we plan to conduct this experiment in the future. Despite this, our current evidence regarding intracellular ROS detected using DCFH-DA and hepatic ROS measured using ELISA still supports our conclusion. We greatly appreciate your helpful suggestion and look forward to future opportunities to further validate our results.

Comment 1.2. Some language errors are still not revised in the revised version, such as "DCFDA" in the main context, and figure legend should be revised as "DCFH-DA".

Response 1.2: We have thoroughly revised the errors of "DCFDA" in the main context and figure legend.

Comment 1.3. In revised Figure 2, the scar bar of the IHC image is missing. And the related problem was also occurred in Figure 4.

Response 1.3: The scar bar of figures has been updated in revised Figure 2 and 4.

Reviewer 2 Report

authrs are required to check inflammation, that is cytokines released in MAFLD treated mice and what was the affect of those cytokines in MAFLD+Berberine treated mice

Author Response

Comment 1.1: Authors are required to check inflammation, that is cytokines released in MAFLD treated mice and what was the effect of those cytokines in MAFLD + Berberine treated mice

Response 1.1: ln our study, we found that Berberine can alleviate hepatic redox imbalance by reducing the levels of ROS and H202. Previous research has shown that an imbalance in the redox state can directly trigger metabolic disorders and inflammation in the liver (Tang C, et al. 2020, doi: 10.1016/j.redox.2020.101519). Our results suggest that Berberine can reduce the release of inflammatory cytokines in mice with MAFLD by improving hepatic oxidative stress. Overall, our findings support the role of Berberine in mitigating liver oxidative stress and reducing inflammation in the MAFLD mice.